# miR-107 Targets NSG1 to Regulate Hypopharyngeal Squamous Cell Carcinoma Progression through ERK Pathway

**DOI:** 10.3390/ijms25115961

**Published:** 2024-05-29

**Authors:** Yifan Hu, Zhizhen He, Baoai Han, Zehua Lin, Peng Zhou, Shuang Li, Shuo Huang, Xiong Chen

**Affiliations:** 1Department of Otolaryngology, Head and Neck Surgery, Zhongnan Hospital of Wuhan University, Wuhan 430071, China; 2022283030074@whu.edu.cn (Y.H.);; 2Sleep Medicine Centre, Zhongnan Hospital of Wuhan University, Wuhan 430071, China

**Keywords:** HSCC, miR-107, NSG1, ERK, targeted therapy, microRNA

## Abstract

Hypopharyngeal squamous cell carcinoma (HSCC) is a kind of malignant tumor with a poor prognosis and low quality of life in the otolaryngology department. It has been found that microRNA (miRNA) plays an important role in the occurrence and development of various tumors. This study found that the expression level of miRNA-107 (miR-107) in HSCC was significantly reduced. Subsequently, we screened out the downstream direct target gene Neuronal Vesicle Trafficking Associated 1 (NSG1) related to miR-107 through bioinformatics analysis and found that the expression of NSG1 was increased in HSCC tissues. Following the overexpression of miR-107 in HSCC cells, it was observed that miR-107 directly suppressed NSG1 expression, leading to increased apoptosis, decreased proliferation, and reduced invasion capabilities of HSCC cells. Subsequent experiments involving the overexpression and knockdown of NSG1 in HSCC cells demonstrated that elevated NSG1 levels enhanced cell proliferation, migration, and invasion, while the opposite effect was observed upon NSG1 knockdown. Further investigations revealed that changes in NSG1 levels in the HSCC cells were accompanied by alterations in ERK signaling pathway proteins, suggesting a potential regulatory role of NSG1 in HSCC cell proliferation, migration, and invasion through the ERK pathway. These findings highlight the significance of miR-107 and NSG1 in hypopharyngeal cancer metastasis, offering promising targets for therapeutic interventions and prognostic evaluations for HSCC.

## 1. Introduction

HSCC is a relatively rare malignant tumor of the head and neck, but it has one of the worst prognoses among head and neck cancers [1]. The early symptoms of this cancer are not typical and are often accompanied by cervical lymph node metastasis, and most patients are already in the advanced stage when the cancer is found, resulting in a low quality of life, poor prognosis, and easy recurrence. The 5-year overall survival rate of this disease is 22~41.8% [2], which is difficult to deal with in head and neck surgery. At present, the main treatment methods for HSCC are surgery combined with chemoradiotherapy, but the therapeutic effect and prognosis of patients are poor [3]. Therefore, finding new treatment options is particularly important. In recent years, targeted therapy has been widely used in clinical practice [4]. The targeted inhibition of oncogene activity prior to surgery has the potential to decrease the tumor volume and limit the extent of tumor invasion, thereby reducing the scope of surgical resection and preserving laryngeal function. This approach presents a viable strategy for enhancing patient prognosis and quality of life, underscoring the importance of identifying the key targets involved in the pathogenesis of HSCC.

The current study suggests that abnormal miRNA expression levels are a key factor in the development of many cancers. For example, Fan et al. found that long non-coding RNA FGD5-AS1 promoted the proliferation of non-small-cell lung cancer cells through the upregulation of FGFRL1 by sponging miR-107 [5]. miR-107 is a member of the miR-15/107 microRNA gene group of microRNAs, which is characterized by the presence of “AGCAGC” in the “seed” region of the first nucleotide or the second nucleotide starting at the 50th end of the mature miRNA [6]. This microRNA is a group of microRNAs with a strong ability to regulate the cell cycle and cell proliferation process, thereby controlling several cell pathways, such as cell proliferation [7] and angiogenesis [8], and participating in the occurrence and development of various diseases, such as neurodegenerative diseases [9] and cancer [10]. Interestingly, despite sharing the same seed sequence, different members of this miRNA family are not equally expressed in cancer cells and may appear to promote or inhibit tumor genesis and development. For example, one study found that miR-107 could be used as an oncogene to promote the progression of gastric cancer [11], and another study on lymphoma found that miR-107 could be used as a tumor suppressor gene to inhibit the tumorigenesis of diffuse large B-cell lymphoma [12]. At present, there are relatively few studies on miR-107 and HSCC, and the effect of miR-107 on promoting or inhibiting tumor development is not clear. The conclusions of this study can fill the relevant gaps in this field.

To further explore the mechanism of action of miR-107 in HSCC, we screened the possible downstream target gene NSG1 of miR-107 through bioinformatics analysis. In previous studies, NSG1 was mainly related to the nervous system, especially the structure and function of nerve cells [13]. However, in recent years, increasingly more studies have been conducted on the relationship between NSG1 and the occurrence and development of malignant tumors. For example, NSG1 expression is significantly increased in the tissues of patients with esophageal cancer and colorectal cancer [14,15]. At present, the role of NSG1 in HSCC is still unclear and is worth exploring.

Finally, to further explore the downstream signaling pathway in which the miR-107/NSG1 axis influences HSCC development, we investigated extracellular signal-regulated kinase 1/2 (ERK1/2), which is an important member of the mitogen-activated protein kinase (MAPK) signaling pathway. Previous studies found that NSG1 can promote the EMT of esophageal squamous cell carcinoma cells by activating the ERK signaling pathway [16], and the ERK1/2 signaling pathway was also found to be abnormally activated in esophageal squamous cell carcinoma, gastric cancer, and pancreatic cancer cells [17,18,19], which can promote the proliferation, invasion, and metastasis of cancer cells.

In this study, we aimed to investigate the functions of miR-107 and NSG1 in HSCC cells and their potential mechanisms of action. The results showed that miR-107 was downregulated in HSCC tissues and NSG1 was upregulated in HSCC tissues compared with neighboring normal tissues. In addition, the overexpression of miR-107 can inhibit the proliferation, migration, and invasion of HSCC cells and promote cell apoptosis by downregulating NSG1 and inhibiting ERK pathway signaling. The knockdown and overexpression of NSG1 alone also inhibited and promoted the proliferation, migration, and invasion of HSCC by acting on the ERK pathway. Therefore, our study suggests that miR-107 and NSG1 are targets that play a key role in the development of HSCC and may be used as targeted therapeutic sites for HSCC in the future.

## 2. Results

### 2.1. Decreased miR-107 Expression and Upregulation of NSG1 in HSCC Tissue and Cells

To investigate the clinical relevance and pro-cancer effect of miR-107 in HSCC, we collected 11 HSCC tumor tissue samples and matched them with adjacent non-malignant hypopharyngeal tissue for a total of 22 cases in 11 pairs. Among them, five pairs of specimens were used for RNA extraction, and six pairs of specimens were used for protein extraction. We detected the expression level of miR-107 through RT-qPCR. The analysis showed that compared with the adjacent tissues, the miR-107 expression was abnormally downregulated in cancer tissues (Figure 1A). In addition, we detected the expression levels of miR-107 in multiple HSCC cell lines. It was evident that the expression levels of miR-107 in the FaDu, D562, TU212, and TU686 cell lines were significantly lower than those in the human normal epithelial (HBE) cells (Figure 1B). Furthermore, we predicted the downstream target genes of miR-107 from three databases: miRDB, miRTarBase, and TargetScan. After merging the duplicate results, we identified 84 candidate genes. Validation using miRNA and mRNA expression data from TCGA for HNSCC revealed that NSG1 negatively correlated with the miR-107 expression (Figure 1C). Furthermore, we used TargetScanHuman 8.0 to predict the NSG1 mRNA 3 ‘UTR sequence in combination with miR-107 (Figure 1D).

Subsequently, we assessed the NSG1 expression in the obtained cancer and adjacent non-malignant hypopharyngeal tissue through qPCR, Western blot, and immunohistochemistry. Our findings revealed a significant increase in the NSG1 expression in the HSCC tissue (Figure 2A). Furthermore, we examined the expression level of NSG1 in the HSCC cell lines using Western blot and observed a marked elevation in NSG1 expression compared with the HBE cells (Figure 2B). The relative expression levels of NSG1 in HSCC tissue (*n* = 6) and adjacent non-malignant hypopharyngeal tissue (*n* = 6) were then determined using immunohistochemical methods, confirming the high levels of the NSG1 expression in the tumor tissue (Figure 2C). Finally, an additional validation by qPCR was conducted to confirm the relative expression levels of NSG1 in the HSCC tissue (*n* = 6) and adjacent non-malignant hypopharyngeal tissue (*n* = 6), which yielded results consistent with expectations (Figure 2D).

### 2.2. Overexpression of miR-107 Targeted the Downregulation of NSG1 and Promoted the Apoptosis of FaDu Cells, Which Inhibited Their Proliferation and Invasion

To further understand the role of miR-107 in the development of HSCC, we constructed a FaDu cell line that overexpressed miR-107 using a lentivirus carrying miR-107 and validated the overexpression effect through qPCR experiments (Figure 3A). Subsequently, we found a significant decrease in NSG1 expression in the overexpressed FaDu cells (Figure 3B). Through Transwell experiments, we also found a significant decrease in the cell invasion ability after the transfection (Figure 3C), and CCK-8 experiments showed a significant decrease in proliferation activity of transfected cells within 5 days (Figure 3D). In addition, we used flow cytometry to detect the apoptosis rate of FaDu cells and found that under the same culture conditions, the apoptosis rate of cells that overexpressed miR-107 was significantly increased (Figure 3E).

### 2.3. Knocking down NSG1 Inhibited the Proliferation, Migration, and Invasion Ability of the HSCC Cells

To verify the role of NSG1 in invasion of the HSCC cell lines, we used two high-expression NSG1 HSCC cell lines, namely, FaDu and D562, in the following experiments. We used two different knockdown sequences, namely, Sh-NSG1-1 and Sh-NSG1-2, to knock down NSG1 in the cell lines. Protein blotting confirmed that the expression of NSG1 decreased by more than 65% after the knockdown. In the next experiment, we used cells from the sh2 group (Figure 4A). The CCK-8 experiment showed that silencing NSG1 significantly weakened the survival ability of the cancer cells and reduced the cell viability (Figure 4B). The plate-cloning experiment also showed that after silencing NSG1, the clonal proliferation ability of cells significantly decreased (Figure 4C). The results of the wound-healing experiment showed that after silencing NSG1, the migration abilities of the FaDu and D562 cells were significantly reduced (Figure 4D). The results of the Transwell invasion experiment showed that after silencing NSG1, the invasion abilities of the FaDu and D562 cells were significantly reduced (Figure 4E). All the above experiments were validated in two cell lines and were independent experiments.

### 2.4. Overexpression of NSG1 Promoted the Proliferation, Migration, and Invasion of Hypopharyngeal Cancer Cells

To further validate the role of NSG1 in the invasion of head and neck squamous cell carcinoma cell lines, we overexpressed NSG1 in the HSCC cell lines FaDu and D562 and observed its phenotype. First, we validated the efficiency of overexpressing NSG1 through Western blot, and the results show a significant increase in NSG1 expression after the overexpression (Figure 5A). The CCK-8 experiment showed that the overexpression of NSG1 enhanced the proliferation ability of cancer cells and reduced the cell viability (Figure 5B). Plate-cloning experiments also showed that the overexpression of NSG1 significantly increased the proliferation abilities of the cells (Figure 5C). In addition, it was found through scratch and Transwell invasion experiments that the overexpressed NSG1 showed stronger migration and invasion abilities compared with the control group cells (Figure 5D,E). All the above experiments were validated in two cell lines and were independent experiments.

### 2.5. miR-107 Targeted NSG1 to Affect the Invasion and Cell Viability of HSCC Cells by Regulating the ERK Pathways

To discover the downstream mechanism of NSG1 that regulates the growth of pharyngeal cancer cells, we observed the expression of ERK pathway proteins in the treated cells. It was found that after the overexpression of miR-107, the expression of the ERK pathway active protein P-ERK decreased, while the expression of ERK remained almost unchanged. Similarly, the same effect was observed in the silenced NSG1 HSCC cell lines FaDu and D562. In contrast, we found an increase in the expression of active P-ERK in the HSCC cells that overexpressed NSG1 (Figure 6A). In addition, to further verify whether the ERK pathway could truly promote the proliferation, invasion, and migration abilities of nasopharyngeal cancer cells, we added the ERK pathway agonist H-Ile Lys Val Ala Val OH (concentration of 0.02 mmol/mL) to the cells and validated its effect using Western blot experiments (Figure 6B). Through flow cytometry detection, it was found that the apoptosis rate of the cells significantly decreased with the addition of the ERK pathway activators (Figure 6C). The Transwell experiment found that the addition of ERK pathway agonists significantly increased the cell penetration, which indicates that the activation of the ERK pathway could enhance the cell invasion ability (Figure 6D). The CCK-8 experiment showed a significant difference in cell viability between the control group and the cells on day 5 (Figure 6E).

### 2.6. Overexpression of miR-107 Weakened the Tumorigenic Ability of HSCC Cells In Vivo

In addition, to determine whether the miR-107 could regulate the invasion and migration of tumors in vivo, 12 nude mice were randomly divided into two groups, one of which was the NC group inoculated with empty somatic cells, and the other was the OE-miR group inoculated with overexpressing miR-107 cells. The results showed that compared with the mice in the inoculated NC group, from day 19 to day 24, the tumor formation volume of the nude mice in the OE-miR group was significantly lower than that in the control group (Figure 7A), and the tumor mass was significantly smaller on day 24 after the inoculation (Figure 7B). HE staining showed that the cells of the tumor tissue formed by the NC group had obvious nuclear deep staining, while the tumor tissue morphology of the OE-miR group was closer to the normal tissue (Figure 7C). Immunohistochemical analysis also showed that the expression levels of Ki67 and NSG1 were lower in the tumors formed by the OE-miR group (Figure 7D).

## 3. Discussion

This is the first time we found that miR-107 can regulate the malignant behavior of HSCC cells by inhibiting NSG1. In this study, we found that miR-107 is downregulated in HSCC tissues. In vitro experiments showed that the overexpression of miR-107 could inhibit the proliferation, migration, and invasion of HSCC cells. To find the downstream regulatory pathways, we predicted the possible downstream target gene NSG1 of miR-107 through bioinformatics analysis and verified that miR-107 can inhibit the expression of NSG1 in cells. Next, we investigated the biological effects of NSG1 on the HSCC cells by knocking down and overexpressing NSG1 in the HSCC cell lines. The results showed that NSG1 could promote the proliferation, migration, and invasion of HSCC cells. Because the ERK pathway is a classical pathway for the development of malignant tumors, we also detected the expression of ERK pathway proteins in cell lines that knocked down and overexpressed NSG1, and found that the change trend of P-ERK expression was consistent with that of NSG1. We then added ERK agonists to HSCC cells and verified that the ERK pathway can promote the malignant biological behavior of HSCC cells. Finally, we confirmed through in vivo tumorigenic experiments that the overexpression of miR-107 in HSCC cells can reduce its tumorigenic ability. The detailed mechanism is shown in Figure 8. Therefore, both miR-107 and NSG1 are targets that play vital roles in the development of HSCC and play a regulatory role through the ERK signaling pathway.

miRNA usually binds to the 3′ untranslated or stable region of mRNA and affects the expression of target genes by inhibiting the translation of target genes or promoting the degradation of target genes [20]. Relevant studies showed that miRNA can target the expression of proto-oncogenes or tumor suppressor genes to regulate the biological behavior of tumors [21]. For example, in colon cancer cell lines, miRNA-145-5p can regulate the expression of phosphoserine aminotransferase 1 (PSAT1) and inhibit the proliferation of colon cancer cells [22]. In addition, miR-451 was found to inhibit the epithelial mesenchymal transition (EMT) of glioma cells, as well as its invasion and metastasis [23]. Another study found that miR-10b-5p could promote the invasion and metastasis of gastric cancer cells [24]. The miR-15/107 family is a special group of miRNAs that are highly conserved with a common nucleotide sequence “AGCAGC”, which enables them to recognize several common targets and regulate the cell cycle and cell proliferation [25]. The miR-15/107 group members mainly include miR-15a-5p, miR-15b-5p, miR-16-5p, miR-103a-3p, miR-107, and miR-195-5p [26]. However, it is worth mentioning that different members have different roles in cancer cells. For example, Wang et al. found that miR-103a-3p secreted by cancer-related fibroblasts inhibits apoptosis in NSCLC and promotes cisplatin resistance [27]. Another study showed that miR-195-5p exerts a tumor-suppressive function in human lung cancer cells by targeting TrxR2 [28]. The strong influence of miR-15/107 group members on the proliferation potential of cancer cells makes them ideal candidates for developing novel alternative treatment strategies for cancer based on miRNA [29]. Previous studies showed that miR-107 may be involved in the occurrence and development of hepatocellular carcinoma [30,31], colorectal cancer [32], and gastric cancer [33], and even in the autophagy process of epidermal keratinocytes [34]. This suggests that it is a key gene with a big influence. However, the research on the effect and mechanism of miR-107 in HSCC progression is relatively limited. In this study, we first observed the abnormal downregulation of miR-107 in HSCC tissues and verified it in four HSCC cell lines with consistent results. It is worth mentioning that the change trend of the apoptosis rate caused by miR-107 was the same as that of the cell invasiveness. Although apoptosis could affect the cell invasion result, the results show that the cell invasiveness change trend was significantly greater than that of the cell apoptosis rate under the same conditions, and thus, we believe that miR-107 can affect cell invasiveness. This means that miR-107 can be used as a potential site for the targeted therapy of HSCC and has clinical application value.

To further explore the downstream regulatory pathway of miR-107, we found that NSG1 was a possible downstream target gene of miR-107 through bioinformatics analysis. One study found that NSG1-IgG was elevated in the serum of patients with early-stage colon cancer and was a predictive marker [14]. Another study found that NSG1 can promote the malignant behavior of esophageal cancer cells and can activate the ERK signaling pathway [16], which also provides information for our subsequent exploration of signaling pathways. By overexpressing and knocking down the NSG1 gene in HSCC cell lines, we found that NSG1 could enhance the viability, migration, and invasion ability of HSCC cells. NSG1 could also affect apoptosis because its downstream ERK signaling pathway could affect the apoptosis level in cells. Since the enhancement of cell viability, migration, and invasion is closely related to the malignant phenotype of cancer cells, the expression of NSG1 may potentially promote the occurrence and development of HSCC.

The activation of the ERK signaling pathway is closely related to the occurrence and development of various malignant tumors. The activation of the ERK signal can promote the proliferation, metastasis, and migration of various malignant tumor cells, such as intrahepatic cholangiocarcinoma [35], non-small cell lung cancer [36], and gastric cancer [19]. There have also been many studies on the occurrence and development of the ERK signaling pathway and HSCC. For example, in Wang’s study, FUT6 regulated HSCC proliferation, migration, and invasion by regulating EGFR/ERK/STAT signaling pathways [37]. Granda et al. found that destroying the ERK pathway with drugs can eliminate the cancer-promoting effect of miR-301a overexpression in HSCC [38]. All these studies confirmed that the ERK pathway plays an important role in mediating HSCC cell invasion, metastasis, and proliferation. In our study, we found that the overexpression of miR-107 inhibited the P-ERK expression while decreasing NSG1. We then demonstrated in the FaDu and D562 cell lines that the overexpression of NSG1 promoted P-ERK expression, while silencing NSG1 suppressed P-ERK expression. To further verify the role of the ERK signaling pathway on HSCC cells, we used the ERK pathway agonist H-Ile Lys Val Ala Val OH on the FaDu cells. We found that the ERK agonist H-Ile Lys Val Ala Val OH could promote the proliferation and invasion ability of FaDu cells while inhibiting apoptosis. These findings suggest that the miR-107/NSG1 axis can indeed alter the migration, proliferation, and invasion of HSCC cells by regulating the ERK signaling pathway.

In recent years, research has shown that miRNAs can be used in the diagnosis and treatment of diseases. Liquid biopsy is a test that identifies signature miRNAs present in biological fluids such as saliva, serum, plasma, and even whole blood to aid in the early diagnosis of oral cancer. Previous studies have shown that certain miR-NAs, such as miR-21 and miR-31, can serve as biomarkers for the early diagnosis of oral cancer [39]. In addition, miRNAs and their target molecules can be used to develop clinically targeted drugs. Currently, two oral cancer therapies based on miR-29b and miR-29 have entered clinical trials (Trial ref No.: NCT02009852 and NCT01927354) [40]. Another study found that overexpression of miRNA-34 in mice via adenovirus could inhibit the occurrence and development of lung adenocarcinoma, demonstrating its potential for clinical application [41]. In terms of clinical application, researchers have tested selective inhibitors of miRNA-221 in treating patients with refractory advanced cancer. Of the seventeen patients who received the experimental treatment, nine showed stabilization or improvement after excluding one invalid data point, while seven patients’ conditions continued to progress. Ultimately, the study determined an appropriate phase II dose of 5 mg/kg. The experimental results show that the selective inhibitor of miRNA-221 has good tolerance, safety, and anti-tumor activity in humans [42]. These findings highlight the significant potential of miRNAs in the diagnosis and treatment of diseases, providing new directions and hope for future clinical applications.

Our study provides a new target for the targeted therapy of HSCC and makes an important contribution to the development of future treatment options. Specifically, our study reveals the potential role of miR-107 and NSG1 in HSCC, laying the foundation for the development of therapeutics against these molecular targets. However, there are still some shortcomings in our research. First, we lack clinical trial data to verify the safety and efficacy of miR-107 and NSG1 in targeted therapies. This means we are currently unable to determine whether these molecular targets can provide the desired therapeutic effect in practical clinical applications and whether they will have adverse effects on patient health. Secondly, due to the limited number of clinical samples collected in this study, we could not fully verify the expression levels of miR-107 and NSG1 in clinical samples. This limitation affects the representativeness and persuasiveness of our findings, thus impacting the general applicability of our conclusions. Looking ahead, we hope to produce safe and effective miR-107-related drugs for the treatment of HSCC through further research and clinical trials. We aim to expand the collection of clinical samples and deepen our understanding of the expression levels and mechanisms of action of miR-107 and NSG1 in different patient groups, to provide more accurate and personalized treatment for HSCC patients. We believe that with continuous efforts, we can achieve this goal in the near future, bringing new hope for the treatment of HSCC.

## 4. Materials and Methods

### 4.1. Bioinformatics Analysis

We utilized three online tools—miRTarBase9.0 (https://mirtarbase.cuhk.edu.cn/, accessed on 22 January 2024), miRDB (https://mirdb.org/, accessed on 21 January 2024), and TargetScan (https://www.targetscan.org/, accessed on 22 January 2024)—to predict the targets of miR-107. The intersection of the results from these three databases was identified as the candidate genes. Head and neck squamous cell carcinoma (HNSCC) miRNA and mRNA expression data were obtained from the TCGA database (https://tcga-data.nci.nih.gov/docs/publications/tcga/, accessed on 17 February 2024). To validate the miRNA–mRNA interactions, we analyzed the inverse correlations between the miRNA expression and mRNA levels. Finally, we used TargetScanHuman 8.0 (https://www.targetscan.org/vert_80/, accessed on 17 February 2024) to predict the NSG1 mRNA 3 ‘UTR sequence with the combination of miR-107.

### 4.2. Patients and Specimens

This study included 11 HSCC specimens and matched them with adjacent non-malignant hypopharyngeal tissue from Zhongnan Hospital of Wuhan University. In addition, each specimen was pathologically diagnosed. When the pathological report showed that there were more normal margins around the cancer tissue, we selected the excised adjacent non-malignant hypopharyngeal tissue as the experimental specimen.

### 4.3. Cell Culture

The FaDu (human pharyngeal squamous), D562 (human pharyngeal carcinoma pleural fluid metastatic), and HBE (human bronchial epithelial) cell lines were from Wuhan Punosai Biotechnology Co., Ltd., while the TU212 (human laryngeal cancer) and TU686 (human laryngeal cancer) cell lines were from Wuhan Qisai Biotechnology Co., Ltd. The FaDu, D562, and HBE cells grew in DMEM high glucose (Servicebio) medium, the TU212 cells grew in IMDM medium (Servicebio), and the TU686 cells grew in RPMI-1640 (Servicebio) medium. We added FBS with a final concentration of 10% to the culture medium before use. Then, we cultivated the cells in a culture incubator under standard conditions. When the cell coverage reached 70% to 85%, passaging and seeding were carried out. In the later-stage experiment, an ERK signaling pathway activator H-Ile Lys Val Ala Val OH was added to the cells (working concentration 0.02 mM/mL, HY-P4322, MCE, Monmouth Junction, NJ, USA).

### 4.4. PCR Assay

Total RNA was extracted with a TRIzol reagent, and cDNA was synthesized according to the procedure of the transcriptase kit (R222-01,Vazyme Biotech, Nanjing, China); incubated at 42, 50, and 85 °C for 2 min, 15 min, and 5 s, respectively; and then stored at 4 °C. Real-time fluorescent quantitative PCR (RT-PCR) reactions were performed on CFX96 Connect (Bio-Rad, Hercules, CA, USA), which used the SYBR Green PCR kit (G3326-01, Ser-vicebio, Wuhan, China) at 95 °C for 30 s, then 95 °C for 15 s, and 60 °C for 30 s. U6 was used as the internal parameter of miR-107, and β-Actin was used as the endogenous control of NSG1. Three RT-PCR reactions were performed. The amplification and dissociation curves of qPCR were evaluated, and the relative expression levels of the target genes were calculated using the 2^−ΔCt^ formula. All primer sequences are shown in Table 1:

### 4.5. Western Blot

First, the cells and tissues were ground and cleaved, and the total protein of the cells was extracted by RIPA lysate with protease and phosphatase inhibitors. The concentration of the protein was determined by a BCA kit (G2026-200T, Servicebio, Wuhan, China) and boiled. The amount of protein in each hole of the ten-hole comb was controlled to be about 20 μg. The proteins were separated by 10% sodium dodecyl sulfate (SDS)-polyacrylamide gel electrophoresis (PAGE) and transferred to a PVDF membrane. The membrane was closed with 5% skim milk for 1 h, and the antibody diluted as required was added and incubated at 4 °C overnight. TBST was used for washing, and HRP IgG was added and incubated at room temperature for 1 h. TBST washed the film again, and the chemiluminescent gel imaging system collected images, where the relative gray value of each target protein band was analyzed by Image J software1.53k (http://imagej.nih.gov/ij, accessed on 8 January 2024), with β-actin and β-tubulin used as internal parameters. By calculating the ratio of the gray value of the target gene to the gray value of the reference gene, the relative expression of the target gene after a different treatment was determined. The following antibodies were used for protein detection: NSG1 (Abs148052, dilution 1:1000; Absin, Shanghai, China), ERK (ab130092, dilution 1:1000; Shanghai, China), P-ERK (abs128832, dilution 1:1000; Absin, Shanghai, China), β-tubulin (AC021, dilution 1:10,000; ABclonal, Wuhan, China), and β-Actin (P30002, dilution 1:2,000; Abmart, Shanghai, China).

### 4.6. Apoptosis Detection by Flow Cytometry

After transfection for 48 h, Annexin-V/PI double staining was used to detect the apoptosis rate. The cells were rinsed 3 times with PBS, and 300 μL of binding buffer was added to resuspend the cells. Next, 10 μL Annexin V-FITC and 5 μL PI (AP101-60, MULTISCIENCES, Hangzhou, China) were added successively, incubated in the dark at room temperature for 15 min, and then the stained apoptotic cells were detected by flow cytometry.

### 4.7. Cell Counting Kit 8

The treated cells and control cells were inoculated in 96-well plates with 1000 cells per well. Fresh medium containing 10% Cell Counting Kit 8 (CCK8) reagent (Biosharp, BS350A, Hefei, China) was replaced and incubated at 37 °C for 1 h, and the OD value was measured at 450 nm.

### 4.8. Cell Transfection

Both the lentivirus silencing and overexpressing plasmids were from Shanghai Jikai. The resuscitated FaDu and D562 cells were cultured in DMEM complete medium in a 5% CO_2_ incubator at 37 °C, and then passed through after fusion. During transfection, the cells of the logarithmic growth stage were collected and incubated at 37 °C and 5% CO_2_ with lentiviral vectors and viral infectious agents. After the lentivirus transfection for 48 h, 2 μg/mL of puromycin was added and screened for 24 h to 48 h. The transfection experiment of each gene in each cell was carried out three times, each time for independent repetition.

### 4.9. Wound-Healing Assay

The cells were scraped with a 200 μL pipette tip and cultured in a serum-free medium (37 °C, 5% CO_2_) for 48 h. The mean migration distance of the cells was recorded under a 100-fold microscope.

### 4.10. Colony Formation

Each well in the 6-well plate was inoculated with 500 treatment cells or 500 control cells. After the cell colony was fixed and visible to the naked eye, it was stained with 0.3% crystal violet in ethanol for 15 min. Finally, pictures of the whole well were taken.

### 4.11. Transwell Invasion Assay

Cell invasion was measured using Matrigel-precoated Transwell inserts (cat. 353097, corning, Becton Dickinson and Company, East Rutherford, NJ, USA). The chambers were pretreated with Matrigel (cat. 354234, Becton, Dickinson and Company, East Rutherford, NJ, USA). Cells were seeded in Transwell inserts at a concentration of 1 × 10^6^ cells per well and incubated for 48 h. Each assay was repeated in triplicate.

### 4.12. Animal Study

Twelve 4-week-old male nude mice were fed in the Animal Center of Zhongnan Hospital of Wuhan University and randomly divided into two equal-sized groups. Each mouse was inoculated with 5 × 10^6^ FaDu cells subcutaneously, while one group was inoculated with the control FaDu cells and the other group was inoculated with the FaDu cells that overexpressed miR-107. During the reproduction in the mice, the tumor volume was recorded every 3 days, and the calculation formula was a × b × b × 0.5 (a was the long axis, b was the short axis). All mice were killed and weighed after carbon dioxide inhalation and cervical dislocation. The tumor was removed from the subcutaneous tissue and weighed. The tumor tissues were fixed with 4% paraformaldehyde, embedded in paraffin, and sliced for HE staining and immunohistochemical analysis.

### 4.13. Hematoxylin and Eosin Staining

Hematoxylin and eosin (HE) staining was undertaken with an HE staining kit (G1076, Servicebio) and pathological changes were observed (400× magnification).

### 4.14. Immunohistochemistry

Tumor tissues were fixed and dehydrated using an ethanol gradient. Then, immunohistochemistry (IHC) staining was conducted by following the instructions of the immunohistochemical kit (36311ES50, Yeasen, Shanghai, China). Anti-Ki-67 antibodies (dilution 1:500; cat. GB111141,100, Service-bio, Wuhan, China) and Anti-NSG1 antibodies (dilution 1:1000; abs148052, Absin, Shanghai, China) were incubated overnight at 4 °C. Finally, Image J v1.8.0 (National Institutes of Health, Bethesda, MA, USA) was used for the analysis.

### 4.15. Data Analysis

Data are expressed as the mean and standard deviation (mean ± SD) of at least three result replicates. To calculate the statistical probabilities, unpaired Student T-tests or one-way ANOVA were performed where appropriate. Statistical analysis and mapping were performed using GraphPad Prism 9.5. *p* < 0.05 was considered to be statistically significant.

## 5. Conclusions

In summary, miR-107 can regulate NSG1 to affect the malignant behavior of HSCC cells by inhibiting the ERK signaling pathway. In addition, NSG1 expression was elevated in HSCC, which may have been involved in the occurrence and development of HSCC. Currently, siRNA- and miRNA-based therapies have entered clinical trials [43], and their application prospects are promising. Therefore, miR-107 and NSG1 are promising molecular targets for the diagnosis and treatment of HSCC, and their application in targeted therapy is of great significance in improving the survival rate and prognosis of HSCC patients.

## Figures and Tables

**Figure 1 ijms-25-05961-f001:**
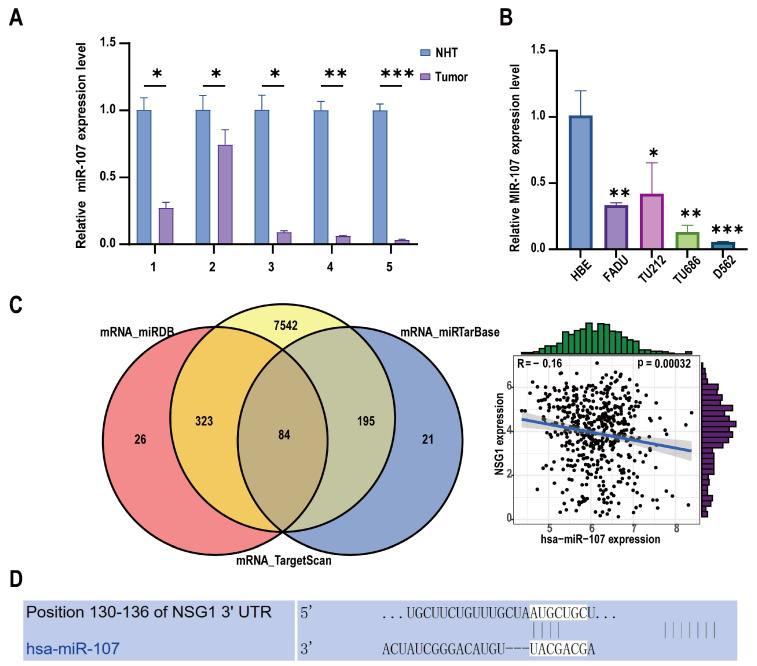
Expression levels of miR-107 in HSCC tissues and cells. (**A**) The relative expression levels of miR-107 in the HSCC tissues (*n* = 5) and adjacent tissues (*n* = 5) were detected using RT qPCR. (**B**) We detected the relative expression levels of miR-107 in the HSCC cells through RT qPCR. (**C**) A total of 84 predicted downstream genes were screened through the database, and the expression trends of NSG1 and miR-107 were opposite to each other. (**D**) The predicted binding sequence diagram of NSG1 mRNA 3′ UTR to miR-107. * *p* < 0.05, ** *p* < 0.01, *** *p* < 0.001. Relative expression level refers to the ratio of the expression of the target gene to the expression of the internal reference gene under the same condition. This meaning is the same in the following chart.

**Figure 2 ijms-25-05961-f002:**
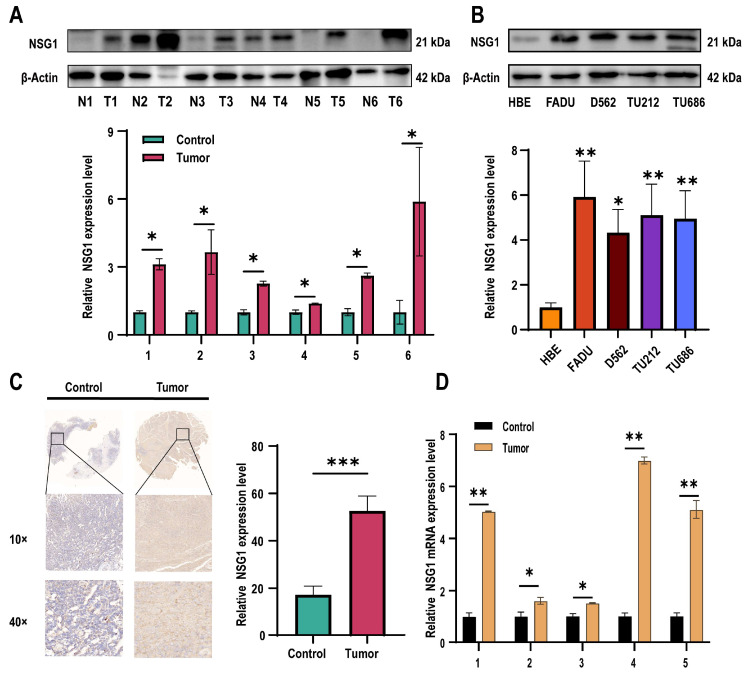
Expression levels of NSG1 in HSCC tissues and cells. (**A**) The relative expression levels of NSG1 in the HSCC tissue (*n* = 6) and adjacent non-malignant hypopharyngeal tissue (*n* = 6) were detected by Western blot. (**B**) Western blot was used to detect the relative expression level of NSG1 in the HSCC cells. (**C**) We detected the relative expression levels of NSG1 in the HSCC tissue (*n* = 6) and adjacent non-malignant hypopharyngeal tissue (*n* = 6) using immunohistochemical methods. (**D**) The relative expression levels of NSG1 in HSCC tissue (*n* = 6) and adjacent non-malignant hypopharyngeal tissue (*n* = 6) were detected by RT qPCR. The data from three independent experiments are expressed as mean ± standard deviation. * *p* < 0.05, ** *p* < 0.01, *** *p* < 0.001.

**Figure 3 ijms-25-05961-f003:**
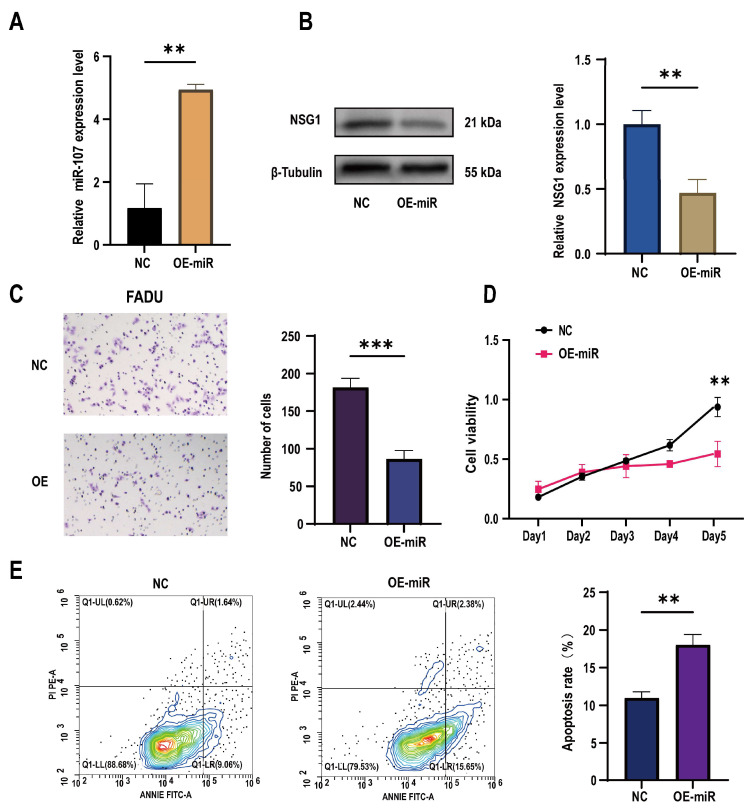
Effects of the overexpression of miR-107 on the apoptosis, proliferation, and invasion of the FaDu cells. (**A**) The FaDu cells were transfected with miR-107-loaded lentivirus and negative control lentivirus (NC), and RT-qPCR was used to verify the expression of miR-107 after the transfection. (**B**) Western blot analysis was performed to detect the expression of NSG1 in the transfected cells. (**C**) The invasion ability of the transversal FaDu cells was assessed by a Transwell assay (200×). (**D**) The proliferation of the overexpressed miR-107 cells was measured by the CCK-8 method. (**E**) The apoptosis of the transfected cells was detected by flow cytometry. Red means more cells; blue means fewer cells. NC group: transfected with miR-107-overexpressed lentiviral empty vector; OE-miR group: transfected with miR-107-overexpressed lentivirus. Data from three independent experiments are expressed as mean ± standard deviation. ** *p* < 0.01, *** *p* < 0.001.

**Figure 4 ijms-25-05961-f004:**
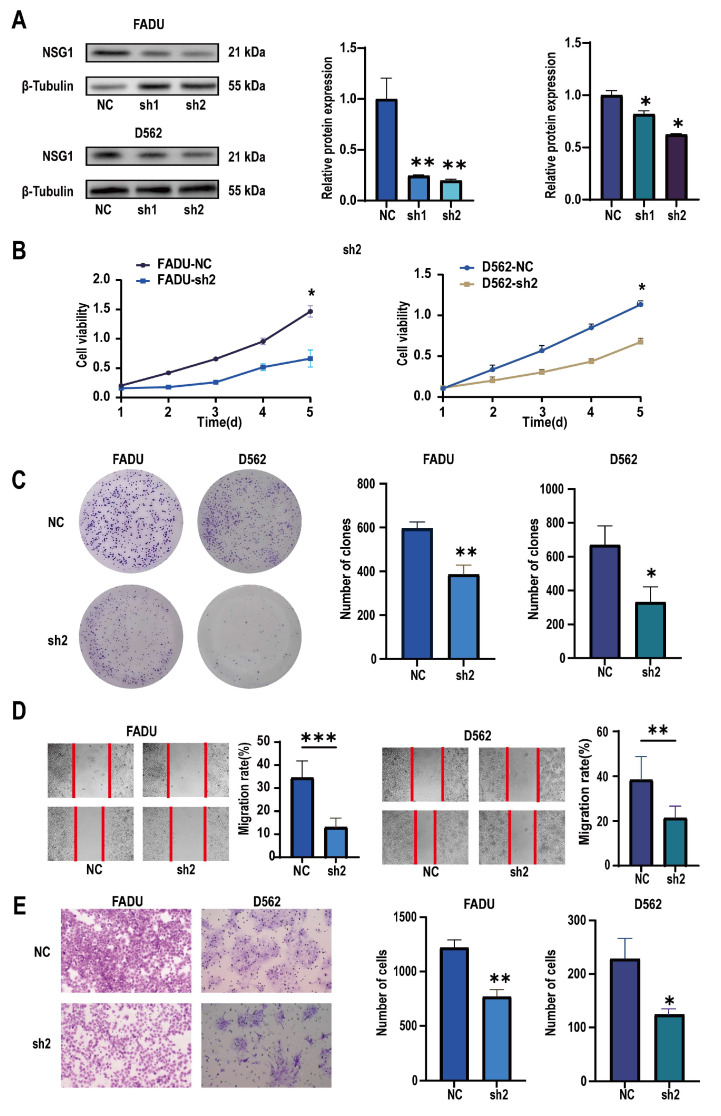
Changes in the proliferation, migration, and invasion abilities of hypopharyngeal cancer cells after downregulating NSG1 (FaDu and D562 cell lines were transfected with lentiviral vectors containing a control or knockdown NSG1 gene). (**A**) Western blot analysis was performed to detect the expression of NSG1 in the transfected cells. (**B**) The proliferation of NSG1 cells within 5 days after the knockdown was measured by the CCK-8 method. (**C**) The proliferation capacity of the HSCC cells was detected by the colony formation method. (**D**) The migration ability of transfected cells was tested by a wound scratch assay. (**E**) The invasion ability of the HSCC cells was evaluated by a Transwell assay. NC group: transfected with an sh-NSG1-NC plasmid; sh1 group: transfected with an sh-NSG1-1 plasmid; sh2 group: transfected with an sh-NSG1-2 plasmid. Data from three independent experiments are expressed as mean ± standard deviation. * *p* < 0.05, ** *p* < 0.01, *** *p* < 0.001.

**Figure 5 ijms-25-05961-f005:**
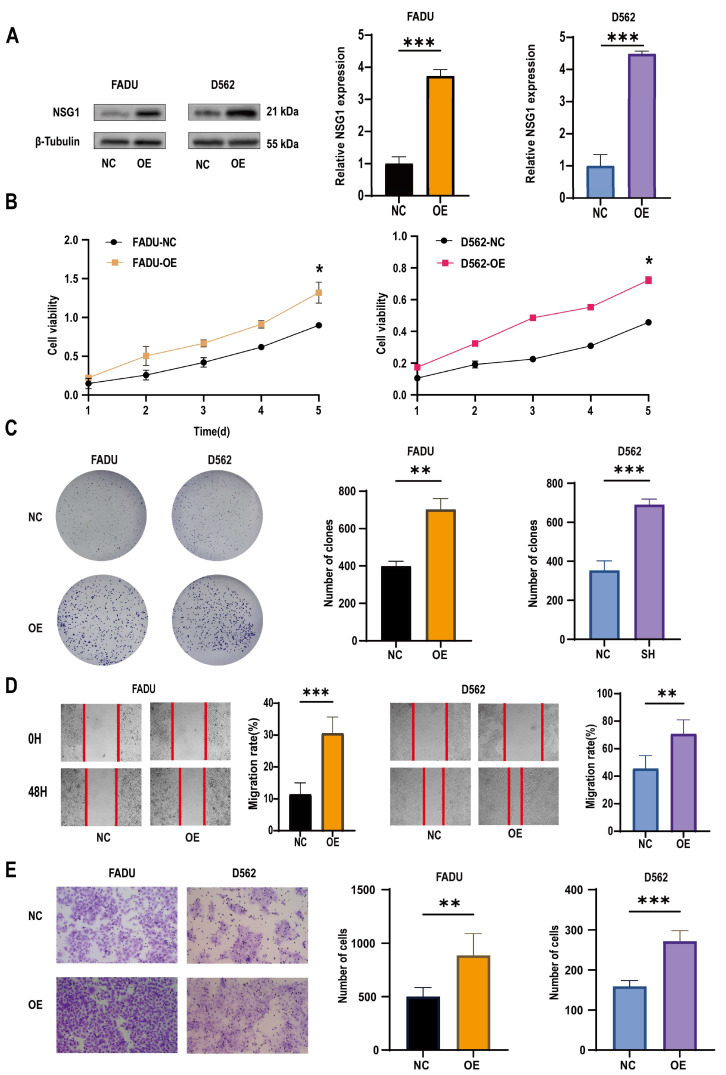
Changes in the proliferation, migration, and invasion ability of HSCC cells after the overexpression of NSG1. (**A**) Western blot was used to detect the expression of NSG1 in transfected cells. (**B**) The CCK-8 method was used to measure the proliferation of cells that overexpressed NSG1 within 5 days. (**C**) A colony formation assay was used to detect the proliferation ability of the NSG1-overexpressing cells. (**D**) We performed wound scratch experiments on the transfected cells to test their migration ability. (**E**) We evaluated the invasive ability of transfected cells through Transwell experiments. NC group: transfected with OE-NSG1-NC plasmid; OE group: transfected with OE-NSG1 plasmid. The data from three independent experiments are expressed as mean ± standard deviation. * *p* < 0.05, ** *p* < 0.01, *** *p* < 0.001.

**Figure 6 ijms-25-05961-f006:**
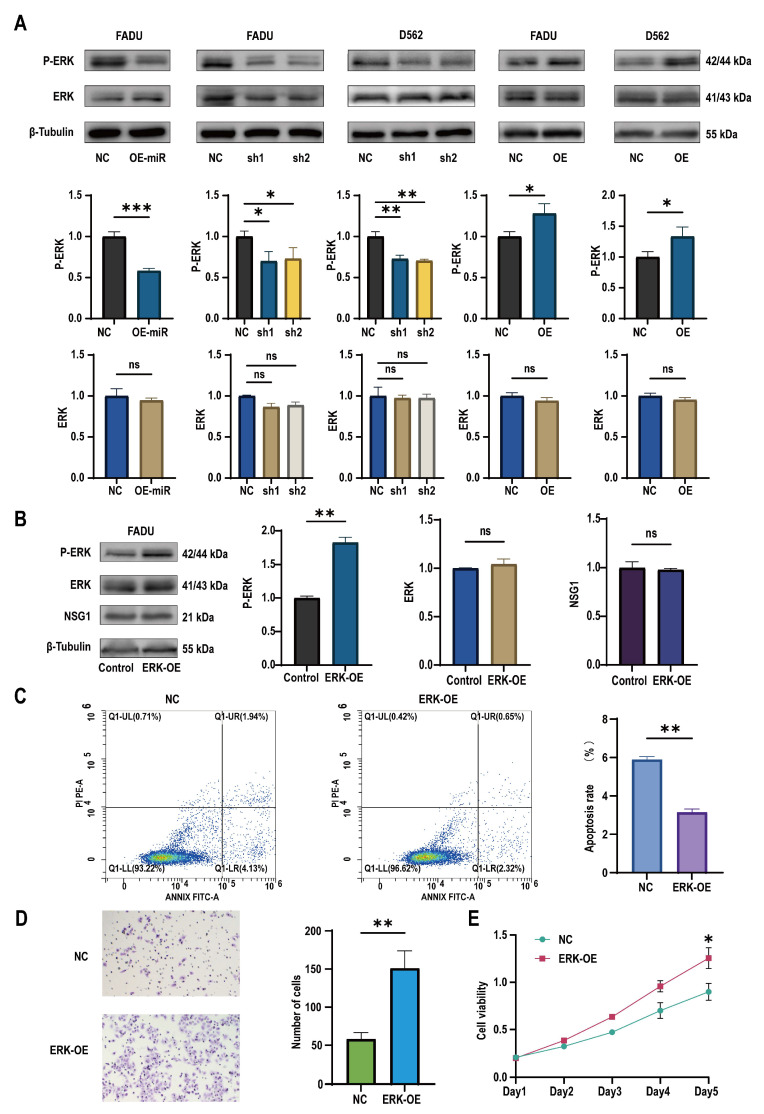
Changes in miR-107 and NSG1 affected the tumor cell phenotype through the ERK pathway. (**A**) Western blot was used to detect the expressions of ERK and P-ERK in the transfected cells. (**B**) Western blot was used to detect the expressions of ERK, P-ERK, and NSG1 in the cells after the addition of the ERK agonists. (**C**) Flow cytometry was used to detect the effects of the ERK agonists on cell apoptosis. Red means more cells; blue means fewer cells. (**D**) We evaluated the invasive ability of cells after adding the ERK agonists through Transwell experiments. (**E**) We measured the proliferation of cells within 5 days after the addition of the ERK pathway agonists using the CCK-8 method. The data from three independent experiments are expressed as mean ± standard deviation. NC group: blank control group; ERK-OE group: added H-Ile Lys Val Ala Val OH group. Other groups’ meanings are the same as in the above chart. ns indicates no statistical significance, * *p* < 0.05, ** *p* < 0.01, *** *p* < 0.001.

**Figure 7 ijms-25-05961-f007:**
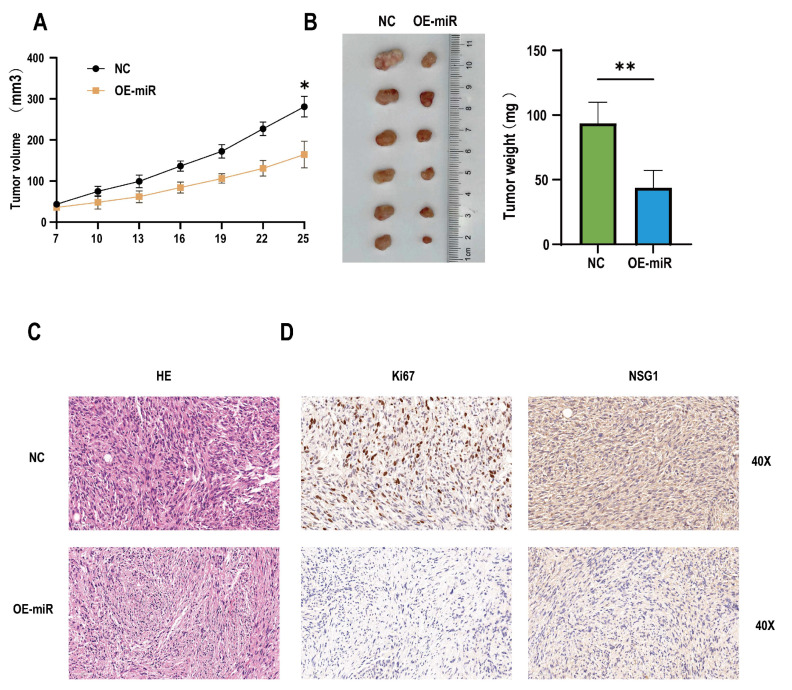
Overexpression of miR-107 inhibited the tumorigenic growth of the hypopharyngeal cancer cells in vivo. (**A**) The tumor volume was measured every 3 days when the tumor was visible to the naked eye. (**B**) On day 25, the tumors were collected and weighed. (**C**) HE staining of the tumor tissue. (**D**) Immunohistochemical staining of Ki-67 and NSG1 was performed in the tumor tissues. Scale = 50 µm. * *p* < 0.05, ** *p* < 0.01.

**Figure 8 ijms-25-05961-f008:**
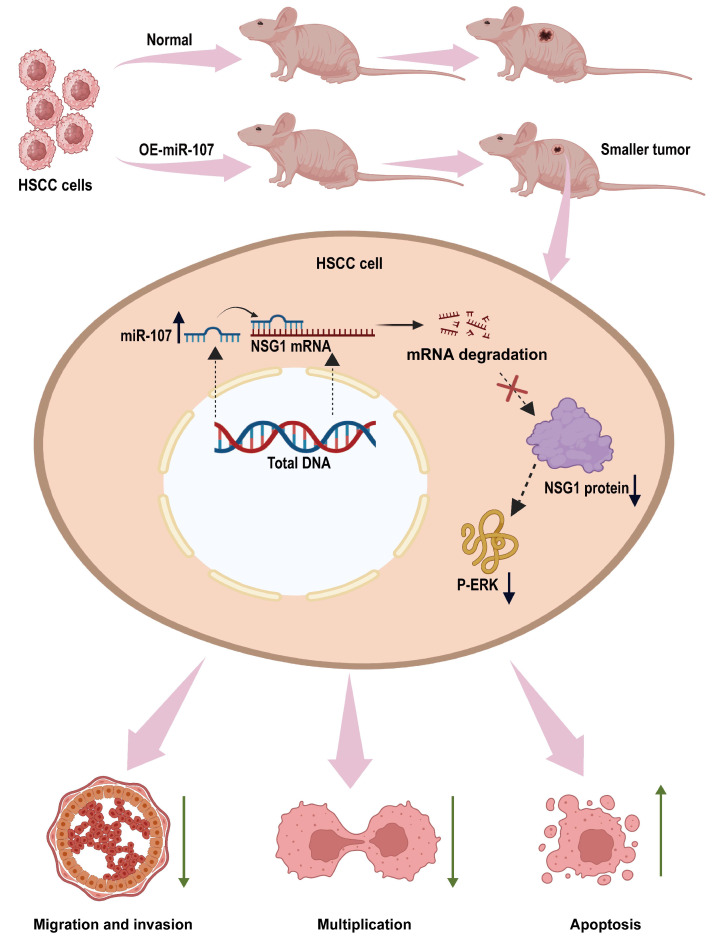
miR-107 targets the NSG1/ERK axis to regulate the HSCC cell invasion, metastasis, and proliferation. miR-107 promotes the apoptosis of HNSC cells and reduces their proliferation, migration, and invasion abilities by inhibiting the expression of NSG1, thereby inhibiting the ERK signaling pathway.

**Table 1 ijms-25-05961-t001:** Primer sequences.

miR-107-RT	GTCGTATCGACTGCAGGGTCCGAGGTATTCGCAGTCGATACGACTGATAG	
	Forward	Reverse
**miR-107**	GCAGCAGCATTGTACAGGG	ACTGCAGGGTCCGAGGTATT
**NSG1**	GTTGGGGAACAATTTCGCAG	GGTGACACCCTCCGTGATG
**β-Actin**	AAGTGTGACGTTGACATCCG	GATCCACATCTGCTGGAAGG
**U6**	TGCCCCACATAATGCTACC	TATGTCCGTCTGTGGAAACC

## Data Availability

All data generated or analyzed during this study are included in this published article. The data that support the findings of this study are available from the corresponding author upon request.

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
