# Peer review of "miR-107 Targets NSG1 to Regulate Hypopharyngeal Squamous Cell Carcinoma Progression through ERK Pathway"

_ijms, 2024, doi:10.3390/ijms25115961_

Round 1

Reviewer 1 Report

Comments and Suggestions for Authors

Abnormal microRNA expression is one of the factors contributing to the development of many cancers, including HNCs. Many studies indicate that various microRNAs can be used in early diagnosis and the treatment of cancer. Therefore, the topic raised is fully justified.

The Authors sought to investigate the functions and potential mechanisms of miR-107 in HSCC.

Their results indicate that the level of miR-107 is reduced in HSCC tissues compared to adjacent normal tissues. Moreover, miR-107 overexpression can inhibit activity

proliferation, migration and invasion of HSCC cells and further promote cell apoptosis

regulation of NSG1 and activation of ERK pathway signaling. Therefore, the authors believe that miR-107 and NSG1 may serve as potential therapeutic targets for HSCC. 

Studies were carried out in the FaDu, D562 and HBE cell lines and animals (mice). 

The research performed was well documented in 8 figures. 

 References were correctly used both in the Introduction and in the Discussion.

However, I have minor comments

1.     It should have been more emphasized why this particular miRNA was the subject of research.

2.     the spelling of miR should be standardized in the manuscript

3.     The abbreviation NSG1 used for the first time in the text should be expanded.

4.     Do the Authors believe that the presented research has any limitations? 

5.     Wouldn't it be worth checking and confirming the results obtained in the human population in the future?

Author Response

Thank you very much for your comments, which are very helpful to improve the article. Please refer to the attachment for detailed reply.

Reviewer 2 Report

Comments and Suggestions for Authors

The current manuscript titled “MiR-107 targets NSG1 to regulate Hypopharyngeal Squamous Cell Carcinoma Progression through ERK pathway” by Hu and colleagues is an interesting article aiming to discover novel therapeutic targets for HSCC. Briefly, the authors analyzed 11 HSCC and adjacent normal tissues, splitting them for RT-qPCR and western blot. After show that miR-107 is downregulated in HSCC, the authors used 3 target prediction programs (mRNA_miRDB, mRNA_miRTarBase, and mRNA-TargetScan) to identify NSG1 as a putative target of miR-107. As expected, the levels of NSG-1 were upregulated in HSCC compared to normal tissues. The number of samples is small, but the results were consistent across the samples. Next, the authors overexpressed miR-107 in FADU cells, which was associated with an increase in apoptosis and inhibition of both proliferation and invasion. Using a mouse xenograft model, the authors demonstrated that miR-107 overexpression reduces tumor size. The authors also modulated NSG1 levels, both knocking down and overexpressing in FADU and D562 cells, and demonstrated consistently that NSG1 controls proliferation, migration and invasion. Finally, the authors explored ERK1/2 signaling pathway revealing that the phenotypes related to miR-107-NSG1 is dependent on ERK activation.

There are some comments that the authors should address to make the article even stronger.

1. It is difficult to read the manuscript with redundant and convoluted sentences throughout the manuscript. The article needs systematic proofreading.

2. The Material and Methods section lacks details, making the comprehensive understanding of the experiments and results very difficult. Moreover, this section should be sufficiently detailed allowing the reproduction of the experiments with similar outcomes. In this sense, the study is very short. Just a few examples, what was used in cDNA synthesis, oligo-dT or random primers? What were the PCR conditions? In western blot, missing lysis buffer, concentration of total protein loaded in the gel, type of electrophoresis, among many others. Cell transfection is the most critical section. Were the experiments performed one, twice, three times? How many times was the murine xenograft assay performed? Missing the description of the clinical samples. Besides, there is a clear mistake on 4.1. Bioinformatics analysis. This section should be completely revised.

3. The introduction does not adequately bring the problem explored in this study. More specific aspects driving the importance of miR-107 in cancer and in HSCC should be provided, supporting its selection among many others. The same is true about NSG1. The fact that they are poorly explored in HSCCs seems very weak, and not enough. The connection among miR-107, NSG1 and ERK1/2 should clearly be brought to the context of the article. What was the main reason for exploring ERK and not any other signaling pathway?

4. Although the results were consistent, the number of clinical samples is really small. To overcome this shortcoming, I would like to suggest that the authors explore public databases, such as the TCGA, Oncomine, and GEO, to confirm the inverse correlation on miR-107 and NSG1.

5. The authors should state whether the adjacent normal tissues were pathologically inspected, or whether they represent adjacent normal-looking normal tissue. In the scenario of field cancerization, the term adjacent normal-looking oral mucosa seems more appropriate.

6. Comparing Fig. 2C and 2E, we verify that changes in apoptosis rate is quite in the same trend as changes in the invasiveness of the cells. So, is miR-107 killing the cells and not reducing the number of invaded cells?

7. On lines 151-153 (To verify the role of NSG1 in invasion of nasopharyngeal carcinoma cell lines, we used two high expression NSG1 nasopharyngeal carcinoma cell lines FADU and D562 in the following experiments.), the authors bring the idea of nasopharyngeal carcinoma cell lines. FADU is from hypopharynx and D562 is from pharynx (not specified). Do the authors consider nasopharyngeal cells exchangeable with hypopharynx cells? Make this very clear.

8. Cell lines. Although TU212 cells are from hypopharynx, it has been described as a problematic cell line, which is contaminated by other cell lines, TU686 represents a laryngeal squamous cell carcinoma cell line, and the origin of HBE is unknown for this reviewer. Could the authors provide the origin of this cell line? Indeed, describe the origin of each of the cell lines used.

9.The efficiency of the NSG1 downregulation was markedly different between FADU and D652. The downregulation in D562 was quite modest with sequence 1. However, the effects on proliferation, number of colonies, migration and invasion in FADU were similar to D562. In fact, for colony formation, the effects on D562 were even stronger. The authors never make clear or discuss this point. What happened with apoptosis? Is NSG1 knockdown associated with alterations in apoptosis rate?

10. In the discussion, a paragraph with limitations and strength of the study would help the readers to understand the meaningfulness of the results. Furthermore, the future scope of the present study should be discussed.

11. Although the prediction that miR-107 is upstream of NSCG1 was detected in 3 different programs, the luciferase reporter gene experiments are also needed to confirm the relationship between them. If not possible to carry out this assay, this is the type of limitation that should be robustly discussed.

Comments on the Quality of English Language

Many grammatical and typographical errors are present, and an English language review should be undertaken.

Author Response

(The authors gave the same response as above.)

Reviewer 3 Report

Comments and Suggestions for Authors

Dear authors,

The manuscript “MiR-107 targets NSG1 to regulate Hypopharyngeal Squamous Cell Carcinoma Progression through ERK pathway”, ijms-3010909, demonstrates how the miR-107-NSG1 axis affects HSCC progression. The findings could be clinically meaningful and valuable for developing a novel treatment approach for HSCC therapy. It is written in an understandable manner, and with a few major and minor changes, I recommend it for publishing.

Major:

  1. I'm concerned about the adequate selection of control tissues. As written in the manuscript: “we collected 11 HSCC tumor tissue samples and matched them with the corresponding adjacent cancer tissue samples." Please name the type of control tissue samples. Is it adjacent nonmalignant hypopharyngeal tissue (normal hypopharyngeal tissue, NHT) or really cancer tissue samples? If cancer, which one? This should be defined and changed throughout the whole manuscript, including results, figures, and their legends (the term “control” should be replaced with “NHT” and the term “tumor “with “HSCC”).
  2. The miR-107 transcript must be defined. Did you examine miR-107-3p or miR-107-5p transcript? In addition, where possible, this should be specified in the introduction and discussion sections.
  3. Lines 323–325: The section 4.1 Bioinformatic Analysis should be rewritten. The lines provided don't describe the bioinformatic analysis.

Minor:

1.        Why were 5 pairs of specimens used for RNA extraction and 6 pairs of specimens used for protein extraction? Why didn't you extract both RNA and protein from the same samples? If mRNA levels and proteins were taken from the same specimens, the results would be more comparable and useful.

2.        I suggest you use the acronym HSCC instead of Hypopharyngeal Squamous Cell Carcinoma as a key word.

3.        Please define what the term “relative expression level” means and how it was calculated (Fig 1-5)

4.        Line 115 – grammar mistake – instead of “will be detected”, it should be written “was detected”.

5.        Figure 2 presents NSG1 protein expression (not miR-107). Therefore, please correct Fig 2 B explanation accordingly.

6.        Can you explain why NSG1 expression was normalized on b-actin in tissue samples and on b-tubulin in cell culture experiments?

7.        Please uniform the symbols (miR/MiR, p-ERK/P-ERK, ki67/Ki67).

8.        Line 306: The reference is missing.

9.        Sections 4.2, 4.3, and 4.4 require considerable English correction. These sections should be written in the past tense or in passive voice because they discuss the methods used.

10.   Line 338 – As I could understand, the level of RNA expression was normalized only on the endogen control level of expression, and that is 2^-ΔCt, not 2^-ΔΔCt calculation.

11.   What do you mean when you say "ACTB is the internal parameter of NSG1"? What is ACTB? Is ACTB an endogenous control, and which gene is it?

Comments on the Quality of English Language

Minor editing of English language needed

Author Response

(The authors gave the same response as above.)

Round 2

Reviewer 2 Report

Comments and Suggestions for Authors

While the manuscript is now scientifically sound, the English language remains poor throughout. It does not meet the standards set by MDPI journals. I strongly encourage the authors to have this manuscript professionally reviewed to ensure it meets the publication standards.

Comments on the Quality of English Language

While the manuscript is now scientifically sound, the English language remains poor throughout. It does not meet the standards set by MDPI journals. I strongly encourage the authors to have this manuscript professionally reviewed to ensure it meets the publication standards.

Author Response

Dear reviewer,
Thank you for your comprehensive review and valuable comments. We are pleased to hear that you consider the manuscript scientifically sound. We understand your concerns about the quality of the English language. We have arranged for MDPI to edit our manuscript. After a professional language review, we made changes to the article as requested by the MDPI language editor. Thank you again for your valuable suggestions to improve our article. Please refer to the manuscript in detail for specific changes.

Reviewer 3 Report

Comments and Suggestions for Authors

Dear authors,

The manuscript has been sufficiently improved, and with a few minor corrections, I recommend it for publication:

1.      line 385-386 – I suggest you use the term “endogenous control” instead of “internal parameter”

2.      line 387 – The delta-delta Ct method, also known as the 2–∆∆Ct method, is a simple formula used in order to calculate the relative fold gene expression of samples. The two delta symbols stand for the difference in the difference in Ct values between the target and reference gene (endogenous control) under control (NMT) and experimental conditions (cancer) https://pubmed.ncbi.nlm.nih.gov/11846609/, https://toptipbio.com/delta-delta-ct-pcr/

Therefore, the amount of target gene, normalised only to the endogenous reference gene is given by the relative amount of target ∆CT, where ∆Ct = Ct (gene of interest) – Ct (housekeeping gene) and the relative expression level of target gene is 2^-(∆Ct). And the amount of target gene (cancer), normalised to the endogenous reference and relative to a reference sample (NMT), is given by the fold change of target, ∆∆CT, where ∆∆Ct = ∆Ct (test sample/cancer) – ∆Ct (control/NMT) so the fold change of the gene is 2^-(∆∆Ct).

In other words, in the ΔΔCt method of qPCR data analysis, the Ct values obtained from two different experimental RNA samples are directly normalised to a housekeeping gene and then compared. Therefore, firstly, the difference between the Ct values (ΔCt) of the gene of interest and the housekeeping gene is calculated for each experimental sample. Then, the difference in the ΔCt values between the experimental and control samples ΔΔCt is calculated. The fold-change in expression of the gene of interest between the two samples (cancer and NMT) is then equal to 2^(-ΔΔCt).

The RNA expression level in this publication was only normalised to the endogen control level of expression, which is the target gene's relative expression level. This should be calculated as 2^-ΔCt instead of 2^-ΔΔCt. It was computed for both control and cancer samples, and the results are shown in the figures as the relative expression level. Please double-check your calculation and make the necessary corrections.

3.      Figures 4,5,6,7: The explanations of the acronyms should be included to the legends of Figures (SH, SH1, SH2, OE, NC).

Author Response

Dear reviewer,
Thank you very much for your comments. You are the most serious and rigorous reviewer I have ever met, and you have helped me make progress in the process of revising my manuscript. Thank you again for your attention and efforts. I will respond to your three comments in order.
1. We strongly agree with your idea and have revised the article according to your suggestion. The change location can be seen on lines 404-405 of the article.
2. Thank you very much for pointing out my mistakes and guiding me carefully on the difference between the two. I realized that I did make a mistake in understanding these two methods of calculation. I carefully reviewed my calculation process and found that I used the 2^-ΔCt method, but I mistakenly thought it was the 2^-ΔΔCt method. I have modified it in the corresponding part of the article. Thank you again, you let me remember this calculation method forever.
3. Your comments are very reasonable, and we have supplemented them in the explanatory section of the figure. And the article has been polished to further increase the rationality and science of the article. Thank you again for your valuable comments.
Finally, all the revisions can be found in the manuscript, thank you again.